# Genotype-Phenotype Correlations in Neurofibromatosis Type 1: Identification of Novel and Recurrent *NF1* Gene Variants and Correlations with Neurocognitive Phenotype

**DOI:** 10.3390/genes13071130

**Published:** 2022-06-23

**Authors:** Filomena Napolitano, Milena Dell’Aquila, Chiara Terracciano, Giuseppina Franzese, Maria Teresa Gentile, Giulio Piluso, Claudia Santoro, Davide Colavito, Anna Patanè, Paolo De Blasiis, Simone Sampaolo, Simona Paladino, Mariarosa Anna Beatrice Melone

**Affiliations:** 1Department of Advanced Medical and Surgical Sciences, 2nd Division of Neurology, Center for Rare Diseases and InterUniversity Center for Research in Neurosciences, University of Campania “Luigi Vanvitelli”, 80131 Naples, Italy; milena.dellaquila@virgilio.it (M.D.); giuseppina.franzese@unicampania.it (G.F.); simone.sampaolo@unicampania.it (S.S.); 2Neurology Unit, Azienda Unità Sanitaria Locale di Piacenza, 29121 Piacenza, Italy; chiaraterracciano@gmail.com; 3Laboratory of Cellular and Molecular Neuropathology, Department of Environmental, Biological and Pharmaceutical Sciences and Technologies, University of Campania “Luigi Vanvitelli”, 80131 Naples, Italy; mariateresa.gentile@unicampania.it; 4Department of Precision Medicine, University of Campania “Luigi Vanvitelli”, Via L. De Crecchio 7, 80138 Naples, Italy; giulio.piluso@unicampania.it; 5Department of Women’s and Children’s Health and General and Specialized Surgery, University of Campania “Luigi Vanvitelli”, Via Luigi De Crecchio 2, 80138 Naples, Italy; claudia.santoro@unicampania.it; 6Department of Mental and Physical Health and Preventive Medicine, Clinic of Child and Adolescent Neuropsychiatry, University of Campania “Luigi Vanvitelli”, 80131 Naples, Italy; 7R&I Genetics SRL, 35127 Padua, Italy; dcolavito@rigenetics.com (D.C.); apatane@rigenetics.com (A.P.); 8Department of Mental and Physical Health and Preventive Medicine, Section of Human Anatomy, University of Campania “Luigi Vanvitelli”, 80131 Naples, Italy; paolodeblasiis@gmail.com; 9Department of Molecular Medicine and Medical Biotechnology, InterUniversity Center for Research in Neurosciences, University of Naples “Federico II”, 80131 Naples, Italy; spaladin@unina.it; 10Sbarro Institute for Cancer Research and Molecular Medicine, Center for Biotechnology, Temple University, Philadelphia, PA 19122, USA

**Keywords:** Neurofibromatosis type 1, monocentric study cohort, internal phenotypic categorization, *NF1* mutational spectrum, novel and recurrent *NF1* mutations, genotype-phenotype correlations

## Abstract

Neurofibromatosis type 1 (NF1) is one of the most common genetic tumor predisposition syndrome, caused by mutations in the *NF1*. To date, few genotype-phenotype correlations have been discerned in NF1, due to a highly variable clinical presentation. We aimed to study the molecular spectrum of NF1 and genotype-phenotype correlations in a monocentric study cohort of 85 NF1 patients (20 relatives, 65 sporadic cases). Clinical data were collected at the time of the mutation analysis and reviewed for accuracy in this investigation. An internal phenotypic categorization was applied. The 94% of the patients enrolled showed a severe phenotype with at least one systemic complication and a wide range of associated malignancies. Spine deformities were the most common complications in this cohort. We also reported 66 different *NF1* mutations, of which 7 are novel mutations. Correlation analysis identified a slight significant inverse correlation between age at diagnosis and delayed acquisition of psychomotor skills with residual multi-domain cognitive impairment. Odds ratio with 95% confidence interval showed a higher prevalence of learning disabilities in patients carrying frameshift mutations. Overall, our results aim to offer an interesting contribution to studies on the genotype–phenotype of NF1 and in genetic management and counselling.

## 1. Introduction

Neurofibromatosis type 1 (NF1, OMIM #162200), formerly known as von Recklinghausen’s disease, is a complex tumor predisposition syndrome, inherited in autosomal dominant pattern with an estimated incidence of 1:2500–3000 live births [1,2]. The diagnosis of NF1 is based on clinical criteria established by the National Institutes of Health Consensus Development Conference in 1987 and recently updated [3,4]. Inclusion criteria consist on the presence of 6 or more café-au-lait macules (CALMs) over 5 mm in greatest diameter in pre-pubertal individuals and over 15 mm in greatest diameter in post-pubertal individuals, cutaneous or subcutaneous neurofibromas, plexiform neurofibromas (PNFs), axillary or inguinal freckling, optic gliomas, distinctive osseous lesions, two or more iris Lisch nodules and a first-degree relative affected by NF1 [4,5]. Two of these criteria are necessary for clinical diagnosis [3]. Despite NF1 is a completely penetrant disease in adulthood (close to 100%), many of the features are age-dependent with marked inter- and intra-familial variability and expressivity, and generally in this order: CALMs, axillary lentigines, Lisch nodules, and neurofibromas. About 97% of NF1 patients meet the NIH criteria by the age of 8 years and all do so by the age of 20 years [6]; only 50% of children with sporadic NF1 under the age of 2 years meet a single NIH criterion, which often leads to a delay in diagnosis [6]. CALMs are usually the initial clinical feature of NF1 and may occur at birth or in childhood [6]. Axillary lentigines, which appear in early childhood (usually between 3 and 5 years of age), are usually the second manifestation in NF1 children. Cutaneous neurofibromas usually occur in the prepubertal phase, but can develop at a much earlier age and the increase in size and number coincides with puberty and pregnancy. Plexiform neurofibromas, on the other hand, are typically congenital or very early; they can only be recognized as an enlargement of soft tissue or a patch of skin hyperpigmentation [6]. Tibial dysplasia occurs at birth; optic glioma develops in children under the age of 6 years [6].

Possible multisystemic complications including neurological, cardiovascular, gastrointestinal, endocrine, and orthopedic features and neoplastic conditions have been associated to NF1 [7,8].

Based on epidemiologic studies, cancer incidence in NF1 is approximately four-fold higher than in general population [5,9,10]. About 10% of NF1 patients develop malignant peripheral nerve sheath tumors (MPNSTs), usually arising from plexiform neurofibroma and this is the major cause of poor prognosis [11,12]. Pheochromocytoma, sarcoma, melanoma, breast cancer, leukemia, and gastrointestinal stromal tumors (GISTs) are also frequently associated to NF1 [13,14].

More recently, molecular genetic testing was added to the list of diagnostic criteria [2,4,15].

NF1 is caused by mutations in *NF1* (17q11.2), which encodes neurofibromin, a large guanosine triphosphate GTPase-activating protein (GAP), which acts as a tumor suppressor by regulating RAS GTPase [16,17,18,19,20]. To date, more than 3000 different genetic mutations in the *NF1* have been reported in the Human Gene Mutation Database (HGMD) [21,22]. It is estimated that approximately 50% of all NF1 cases are sporadic, caused by de novo variants usually linked to paternal gonadal mutations [21,23]. Single nucleotide variations (SNVs) and small deletions (20 bp or less) account for more than 70% of currently known mutations [21] and most of *NF1* mutations lead to a synthesis of truncated and non-functional neurofibromin [24]. Although the great effort to finely define the correlation between the clinical phenotype and molecular genotype, so far few correlation studies have been recognized, due to high clinical variability, which is observed even within the same family.

*NF1* whole-gene deletion, affecting approximately 4% of NF1 patients, causes severe form of the disease, characterized by cutaneous neurofibromas earlier in life, development of larger number of tumors, including MPNSTs, more frequent and more severe cognitive abnormalities, somatic overgrowth, large hands and feet, and dysmorphic facial features [25,26,27,28]. 3-bp deletion in *NF1* exon 17 (c.2970_2972delAAT) has been associated with typical pigmentary features of NF1 without cutaneous or surface plexiform neurofibromas [29]. *NF1* microdeletions were linked to more severe clinical characteristics and increased lifetime risk for MPNSTs [27,30]. Recently, evidence for a more severe phenotype with higher frequency of spinal and deep neurofibromas were reported in association to the presence of missense mutations located in codons 844–848 of the *NF1* [30]. Missense mutations affecting residue p.Arg1809 in *NF1* were associated to a mild phenotype characterized by multiple café au lait spots, learning disabilities, short stature, and pulmonic stenosis but absence of neurofibromas [31,32,33]. Three specific recurrent non-truncating NF1 hotspots (residues p.Met1149, p.Arg1276, and p.Lys1423) have been linked with Noonan-like features with specific phenotypes: p.Arg1276 and p.Lys1423 pathogenic missense variants were associated with a high prevalence of cardiovascular abnormalities, including pulmonic stenosis; additionally, p.Arg1276 had a high prevalence of symptomatic spinal neurofibromas; p.Met1149 positive cause a mild phenotype, characterized mainly by pigmentary manifestations [34]. In a recent NF1 single-center study, missense variants negatively correlated with neurofibromas while skeletal defects were linked to frameshift variants and whole gene deletions [35]. The c.3721C>T, p.R1241* variant was associated to structural brain alterations, whereas the c.6855C>A, p.Y2285* variant correlated with a higher prevalence of Lisch nodules and endocrinological disorders [35].

The aim of the present study is to delineate the mutational spectrum of *NF1* mutations and to elucidate genotype-phenotype correlations through integrated mutational screening and clinical data collection in a monocentric cohort of selected NF1 patients. We analyzed the *NF1* mutations for their type, pathogenicity, and distribution in the *NF1* and in the domains of the neurofibromin. A genotype-phenotype correlation was performed dividing the clinical features of the NF1 disease into cardinal signs and complications, adopting an internal classification in five NF1 phenotypic groups. Our findings should provide a simple and effective strategy for early diagnosis and genetic counseling in the NF1 disease.

## 2. Materials and Methods

### 2.1. Patients

We retrospectively evaluated the clinical data from a cohort of 420 unselected output patients, referred to our Neurofibromatosis Center at 2nd Division of Neurology of the ‘Luigi Vanvitelli’ University Hospital between January 2013 to December 2020, with a clinical diagnosis or suspicion of NF1. The inclusion criteria consisted of the clinical and/or molecular diagnosis of NF1 during the first clinical evaluation or later during the follow-up (according to the established International Criteria [3,4]).

Our study was conducted on 85 NF1 adult patients [47 females (55%) and 38 males (45%)] matching inclusion criteria. Informed consent was obtained for all patients. NF1 patients were divided into two groups by familial history: familial cases (20 relatives, for a total of 9 NF1 families) and sporadic cases (65 unrelated probands). The clinical data were collected at the time of genetic analysis, updated during follow-up and re-examined for accuracy by clinicians co-authoring this manuscript at the time of this phenotype-genotype correlation study. At least five years follow-up is available for all patients. Clinical and demographic data of the enrolled patients were reviewed retrospectively: sex, age at evaluation, family history, clinical manifestations, as well as *NF1* genetic results, interpretation of variant pathogenicity and mutation site in the NF1 protein domains were evaluated. With regard to the psycho-cognitive phenotype, we used anamnestic-clinical data, sharing previously obtained assessments with standard psychomotor and cognitive tests (e.g., Griffiths’ Scales of Infant Development, GMDS-ER and the Wechsler Scales of Intelligence) according to the patient’s age. To achieve a more stringent genotype/phenotype correlation, the clinical features were divided into cardinal signs and complications, adopting an internal phenotypic categorization into five specific clinical groups. In detail: Group 1 (G1) includes patients with classical NF1 features, presenting with two or more of the following features: 6 or more CALMs, axillary and inguinal freckling, two or more Lisch nodules and neurofibromas, without other manifestations of extra-cutaneous/ocular involvement; Group 2 (G2) encloses NF1 patients presenting with two or more phenotypic features of G1 plus involvement of skeletal apparatus (short stature, scoliosis, hyperkyphosis, bone dysplasia), central nervous system (epilepsy, intracranial vascular malformations, hamartomas/unidentified bright objects/UBOs) and mental system (intellectual disability, anxiety/depression/sleep disorders), vascular system and anomalies of internal organs (heart valve abnormalities and hypertension); Group 3 (G3) includes NF1 patients presenting with two or more phenotypic features of G1, plus multi-apparatus involvement and histological diagnosis of MPNST according to the American Joint Committee on Cancer Staging System for soft tissue sarcomas; Group 4 (G4) includes NF1 patients presenting with two or more phenotypic features of G1 group, plus multi-apparatus involvement and neoplasms of the central (optic glioma, pilocytic astrocytoma) and peripheral (ganglioneuroma, gangliocytoma) nervous systems; Group 5 (G5) includes NF1 patients presenting two or more clinical traits of G1, plus multi-apparatus involvement and neoplasms of variable grade of various organs and apparatus including GIST, endocrine system (pheochromocytomas, thyroid carcinoma), genitourinary system (ovarian, prostatic, testicular, bladder cancer), tumors of the blood series, breast cancer, cutaneous melanoma, ear cholesteatoma.

### 2.2. NF1 Mutation Analysis

Part of NF1 patients clinically evaluated in the present study was previously genetically characterized by RNA analysis and reported by Giugliano et al. [36]. For *NF1* genetic investigation, all DNA was extracted using the Qiagen BioRobot DNA extraction kit (Qiagen Benelux B.V., Venlo, The Netherlands) according to the manufacturer’s instructions and quantified using Nanodrop spectral analysis (Thermo Fischer Scientific, Inc., Waltham, MA, USA). DNA fragmentation and degradation were evaluated by standard agarose gel electrophoresis (100 V, 30 min, 1.5% agarose gel in Tris-borate-EDTA buffer). DNA Library preparation and whole exon enrichment were performed employing Agilent All Exon V.6 kit (Agilent Technologies, Inc., Santa Clara, CA, USA). Library sequences were obtained using the HiSeq2500 Illumina Sequencer (125-bp paired end sequence mode). Bioinformatics analysis included the following: Next-generation sequencing (NGS) reads mapping to whole genomes using the Burrows-Wheeler Alignment tool with default parameters, polymerase chain reaction (PCR) duplicate removal using Picard (http://picard.sourceforge.net, accessed on 31 March 2022), single nucleotide polymorphisms and indel calling using the Genome Analysis Toolkit (GATK) UnifiedGenotyper, variant annotation using snpEff (http://snpeff.sourceforge.net, accessed on 31 March 2022) and false positive variant filtration using the GATK VariantFiltration module. Exome sequencing data and reads alignment analysis were checked for coverage depth and alignment quality employing Bedtools software package. *NF1* variants (RefSeq: NM_000267.3) were filtered for allele frequency (gnomAD minor allele frequency < 1%), considering all potential modes of inheritance and prioritized according to phenotype overlap, gene function, conservation (phyloP) and in silico prediction scores (CADD, SIFT, PolyPhen2, MutationTaster and MutationAssessor). The classification was conducted in accordance with the guidelines from the American College of Medical Genetics and Genomics. In brief, variants were classified as follows: (i) pathogenic variants; (ii) likely pathogenic variants; (iii) variants of uncertain significance (VOUS) [37,38]. Sanger sequencing was performed to validate the presence of significant variants and to perform segregation analysis in related NF1 cases. CNV calling was performed by Varseq software. This algorithm uses changes in coverage depth relative to a collection of reference samples (30 or more reference samples recommended, having on average 100X across all regions, and derived from the same library prep methods) as evidence of CNV events. High-quality CNVs were then annotated and filtered against CNV and gene annotation tracks like OMIM, Orphanet, RefSeq Genes, ClinVar and ClinGen. Real Time PCR was conducted to validate the presence of copy number variations. Briefly, targeted SYBR green fluorescent amplification was performed using primer mapping within exons resulted duplicated or deleted. Threshold amplification cycles were normalized using a housekeeping gene and the resulting values were then compared with the amplification of control samples (ddCt method). Samples with negative CNV bioinformatic analysis were further analyzed by MLPA analysis (MRC-holland, probe set code P081 and P082) according to manufacture instructions. Fragment analysis was performed by Abi3130 automated sequencer equipped with 36 cm capillary array and data analysis was performed using Coffalyser software (MRC-Holland).

### 2.3. Statistical Analysis

Correlation analysis was performed using the Spearman rank correlation test and the point-biserial correlation coefficient for dichotomous variables. In detail, statistical analysis was applied to each of the clinical characteristics common to all NF1 patients in our cohort, to every clinical group (G1-G5) defined in the study, and to the *NF1* mutation type. Odds ratio (OR) with 95% confidence interval (CI) was estimated to evaluate the prevalence of typical clinical symptoms in NF1 patients carrying different types of mutation. p values lower than 0.05 were considered significant.

## 3. Results

### 3.1. NF1 Molecular Findings

Molecular data of enrolled NF1 patients are listed in the Table 1, demographics and clinical features are reported in the Appendix A.

Each NF1 patient was assigned a specific code (NF_ followed by an Arabic number) corresponding to the ‘unique patient identifier’ of the clinical center. Group 1 (G1) includes patients with classical NF1 phenotype, including six or more café-au-lait spots (CALMs), axillary and inguinal freckling, two or more Lisch nodules and neurofibromas, without other manifestations of extra-cutaneous/ocular involvement; Group 2 (G2) encloses NF1 patients presenting the phenotypic features of G1 plus involvement of skeletal apparatus, central nervous system (epilepsy, intracranial vascular malformations, hamartomas/UBOs), and mental system (intellectual disability, anxiety/depression/sleep disorders), vascular system and anomalies of internal organs; Group 3 (G3) includes NF1 patients presenting the phenotypic features of G1 plus multi-apparatus involvement and histological diagnosis of MPNST; Group 4 (G4) includes NF1 patients presenting the phenotypic features of group G1 plus multi-apparatus involvement and neoplasms of the central and peripheral nervous systems; Group 5 (G5) includes NF1 patients presenting the clinical traits of G1 plus multi-apparatus involvement and neoplasms of variable grade of various organs and apparatus. Clinical Significance (CS): P, pathogenic; LP, Likely pathogenic; VOUS, variant of uncertain clinical significance. Family History (FH): Mat, Maternal; Pat, Paternal; Unk, unknown, “-”, sporadic case. Publicly available databases of variants annotated on the disease: ClinVar (www.ncbi.nlm.nih.gov/clinvar, accessed on 31 March 2022), Human Genome Variation Database (HGMD; www.hgmd.cf.ac.uk, accessed on 31 March 2022) and Leiden Open Variation Database (LOVD; databases.lovd.nl/shared/genes). Protein Domain (PD): CSRD, cysteine/serine-rich domain; GRD, GAP-related domain; TBD, tubulin-binding domain; HLR, HEAT-like regions repeat; NLS, nuclear localization signal; Sec14-PH, Sec14 homologous domain and Pleckstrin Homology domain; CTD, C-terminal domain; SBR, Syndecan-Binding Region.

We present a total of 66 *NF1* mutations, including missense, nonsense, start loss, frameshift, splicing, as well as indel mutations. The average depth, depth of the identified variants and minimum and maximum depth obtained by NGS analysis in each sample have been reported in the Appendix A [37,38,105,106].

About 59 mutations (89%) have been previously described elsewhere and 7 are novel mutations (11%). The *NF1* novel mutations are listed below: c.7884_7885del, p.(Phe2629Serfs*9) (NF_8, NF_72); c.4309G>T, p.(Glu1437*) (NF_52); c.4733C>A, p.(Ser1578Tyr) (NF_63); c.5819del, p.(Lys1940Serfs*18) (NF_81); c.6621dup, p.(Trp2208Valfs*13) (NF_85); c.7532C>T, p.(Ala2532Val) (NF_105); c.(4661+1_4662-1)_(7258+1_7259-1)dup (NF_89 and NF_90 related patients). The distribution of all *NF1* mutations identified in our cohort is represented in Figure 1A. By in silico analysis 35 *NF1* variants were predicted as pathogenic (53%), 4 as likely pathogenic (6%), 27 as VOUS (41%). 45 mutations were single base substitutions (68%), including 16 missense (36%), 11 nonsense (24%), 14 splicing (31%), 3 frameshift (7%), 1 start loss (2%). About 10 small base pairs deletions (15%) and 6 duplications (9%) as well as 1 indel mutations were identified. Whole-gene deletions and multi-exons *NF1* duplications/deletions were detected: c.(4661+1_4662-1)_(7258+1_7259-1)dup, c.(586+1_587-1)_(730+1_731-1)del, and c.3497_3974del (Figure 1B).

Of the 85 NF1 patients of our cohort, 65 were sporadic and 20 had a positive family history. Results of *NF1* segregation analysis in the relatives are the follow: c.3G>A, p.? in NF_2, 3, 4; c.1595T>G, p.Leu532Arg in NF_5, 29; c.3826C>T, p.Arg1276* in NF_10, 11; c.3496+1G>A, p.Tyr1106Leufs*28 in NF_23, 25; c.7686delG, p.Ile2563Phefs*40 in NF_31, 32; c.2307dup, p.Thr770Hisfs*6 in NF_45, 46, 47; c.2409+1G>C, p.? in NF_70, 71; c.4278G>C, p.Gln1426His in NF_80, 92; c.(4661+1_4662-1)_(7258+1_7259-1)dup in NF_89, NF_90. Six different *NF1* mutations were shared in unrelated patients: c.1A>G, p.? in NF_28 and NF_48; c.7884_7885del, p.Phe2629Serfs*9 in NF_72 and NF_8; c.4537C>T, p.Arg1513* was common in NF_55 and NF_100; c.1381C>T, p.Arg461* in NF_57 and NF_102; c.4923G>A, p.Trp1641* in NF_73 and NF_26 patients. NF_19 shared the c.3826C>T, p.Arg1276* mutation with NF_10, 11 familial cases. Interestingly, two unrelated patients (NF_56, NF_79) carried two variants in the *NF1*.

### 3.2. Genotype-Phenotype Correlation Study

Genetic and clinical results of all 85 NF1 patients enrolled in this study are summarized in Table 1 and in the Appendix A, respectively. In detail, the 6% showed a mild phenotype and belong to the clinical group G1, with typical NF1 pigmentary manifestations, including CALMs and freckling as well as cutaneous and subcutaneous neurofibromas. The 94% of remaining patients presented at least one systemic complication, often in associations with other tumors, showing a more severe phenotype (G2, G3, G4, G5 clinical groups). Spine deformities, including scoliosis, kyphosis, and joint spondylosis, were the most common complications in our cohort (64/85 patients, 75%). The 83% of patients showed spinal complications. The most common mutations, associated to this clinical trait, were frameshift (15/64 patients, 23.4%) and nonsense (14/64 patients, 23%) variants, potentially resulting in truncated or absent neurofibromin. About 7% of NF1 patients (6/85 patients) presented bone abnormalities (including short stature, osteoporosis, osteopenia, non-ossifying and ossifying fibromas and foot deformities), mainly associated to mutations resulting in a truncated neurofibromin (66%).

In the 50% of NF1 cases showing bone anomalies the *NF1* presented clearly pathogenic variants. 33% of patients manifested both spine and bone deformities (28/85 patients), showing a most severe skeletal phenotype. Nonsense *NF1* mutations were the most common type of variants (34%) associated to this clinical combination (spine and bone alterations). Interestingly, skeletal complications were more common in females than males in our case series (spine deformities: 56% females; bone deformities: 67% females; both phenotypes: 64% females). About 21% of NF1 patients manifested spinal neurofibromas that were strongly associated with pain and scoliosis, with a high prevalence in males (67%).

In this study, cardiovascular anomalies associated to *NF1* mutations (24/85 patients, 28%) included hypertension, aortic and mitral valves insufficiency, Moyamoya disease (MMS), aortic stenosis, and coarctation of the aorta. This phenotype was predominantly evident in patients carrying missense (20%) or nonsense (21%) *NF1* mutations.

UBOs were one of the most frequent brain lesions associated to NF1 [46]. In our cohort 18% of patients (15/85 patients) were affected by UBOs with a higher prevalence in females (12 females, 80%). This clinical condition was associated to different types of variants: missense (27%), nonsense (27%), frameshift (20%), and splicing (7%) mutations.

Less common NF1 complications including obesity (5%, 4/85 patients) and hypercholesterolemia (3/85, 4%) were present in our patients.

PNFs affected 13% of NF1 patients (11/85 patients), 82% of which also presented cutaneous, subcutaneous, or spinal neurofibromas. About 55% (6/11) of these patients showed a more severe clinical phenotype associated to other malignancies (clinical groups G3, G4, G5).

### 3.3. NF1-Associated Malignancies and Other Complications

Malignancies and other non-neoplastic complications may worsen prognosis of NF1 patients.

Nervous system malignancies affected 21% (18/85) of our case series, all grouped in G4. Optic glioma and astrocytoma were the most common for this specific clinical condition (10/85 patients, 12%). About 80% of NF1 patients showing optic glioma or astrocytoma were female, carrying mainly frameshift mutations in the NF1 gene (4/8, 50%). Different types of organ tumors were diagnosed in 19 individuals (19/85 patients, 22%) belonging to G5 clinical group, including GIST (10%), breast (16%), and prostate (5%) cancers, as well as seminoma (5%) and lymphoma (5%). In two cases other malignancies such as ganglioneuroma (NF_37) and optic glioma (NF_65) were associated to NF1. Moreover, three patients of G5 clinical group showed malignant pheochromocytoma/paragangliomas associated to NF1 and carried three different nonsense mutations (p.Arg1276*, p.Trp1641*, and p.Asn2364*) in the *NF1*, distributed in different protein domains (Figure 1A). Interestingly, the p.Arg1276* mutation in the GAP-related domain (GRD) was already reported in one patient with NF1 and pheochromocytoma [77]. Furthermore, the same mutation was recently associated to cardiovascular abnormalities in NF1 disease [34]. Moreover, in our cohort the p.Arg1276* mutation was identified in the related NF_10 and NF_11 patients affected by cardiovascular abnormalities.

The clinical picture of G5 group was worsened to several systemic complications including spine (79%, 15/19 patients), bone (42%, 8/19 patients) deformities and cardiovascular defects (32%, 7/19 patients). Eight patients carried whole *NF1* or large intergenic deletions/duplications associated to different systemic complications, including spine deformities (88%, 7/8 patients), cardiovascular (38%, 3/8 patients), and learning (25%, 2/8 patients) defects. In our cohort five different *NF1* mutations were shared in unrelated patients. c.1A>G, p.? pathogenic mutation was common to NF_28 and NF_48 patients, both female with a similar age, belonging to the G2 clinical group. NF_72 and NF_8 carried the novel mutation c.7884_7885del, p.(Phe2629Serfs*9) located in the syndecan-binding region (SBR) of NF1. Both patients disclosed a severe clinical picture, which in the case of NF_8 the spinal form of NF1 was associated to learning disabilities and obesity, while in NF_72 the disease phenotype was complicated by 25-OH vitamin D deficiency as well as optic glioma and astrocytoma. NF_57 and NF_102 carried the pathogenic mutation c.1381C>T, p.Arg461* and disclosed a clinical phenotype mainly complicated by spine and bone deformities. The nonsense mutation c.4537C>T, p.Arg1513* was shared by NF_55 and NF_100 patients, while the c.3826C>T, p.Arg1276* mutation was shared by NF_19 and two familial cases (NF_10 and NF_11). Both mutations were linked to a complex and severe NF1 phenotype with multisystemic complications including neurological, cardiovascular, endocrine, skeletal, and neoplastic features. Interestingly, in our cohort two unrelated patients (NF_56, NF_79) showed two different variants in the *NF1* gene. In detail, NF_56 carried the c.2409+1G>A, p.? pathogenetic splice site mutation and the c.2375T>A, p.Leu792His likely pathogenetic variant, while NF_79 presented the pathogenic mutation c.3916C>T, p.Arg1306*, and c.1975C>T, p.Arg659Trp classified as VOUS. Unfortunately, for these two patients segregation analysis was not performed as parents or other family members were not available, therefore we cannot disclose the cis or trans location of these mutations. Of interest, both patients disclosed skeletal anomalies as common complications, in association to cardiovascular and gastric disorders in patient NF_56 and tubular adenoma in patient NF_79. Moreover, in our case series a slight but significant inverse correlation between delayed acquisition of psychomotor skills with residual multi-domain cognitive impairment and age at the time of diagnosis (r = −0.35, *p* < 0.001) was observed (Figure 2A). Odds ratio analysis revealed that the probability of having learning disabilities was higher in patients carrying frameshift mutations (OR 6.2, CIs 1.626 to 23.64, *p* < 0.01) (Figure 2B). No other significant correlation between sex, age at diagnosis, clinical groups, single clinical features, and type of mutation was found.

## 4. Discussion

Diagnosis of NF1 is usually based on clinical findings according to NIH diagnostic criteria, nevertheless, owing to the extreme variability in clinical expression and age dependency of most clinical manifestations, molecular testing could represent a simple and effective strategy for early and differential diagnosis. Due to the high spectrum of genotypic and phenotypic heterogeneity few genotype–phenotype correlations have been previously reported [29,30,31,33,34,71]. In this study, we analyzed mutational spectrum of 85 clinically well-characterized familial/sporadic NF1 patients and statistically evaluated prevalence of specific clinical features associated to NF1 in the different *NF1* mutation subtypes. For genotype–phenotype correlation study we have adopted a detailed internal clinical dissection, dividing the NF1 patients in five clinical groups according to disease phenotype. As this is a monocentric study, we reduced the potential bias in clinical evaluation between different centers. NF1 is characterized by complete age-dependent penetrance. The careful selection of the NF1 patients and the availability of the long follow-up, of at least 5 years for each patient, allowed us to have a full clinical view of the NF1 disease and to get insight the possible genotype–phenotype correlations of the *NF1* mutations identified in our investigation.

Our findings showed that the 6% of study cohort disclosed a relatively mild phenotype with typical NF1 pigmentary manifestations (clinical group G1) and the remaining 94% disclosed a more severe phenotype, presented at least one systemic complication (clinical groups G2, G3, G4, G5), and of these the 47% were affected by other malignancies. In this study, we reported 66 different *NF1* mutations, 7 of which were novel mutations. Mutations were distributed along the entire *NF1*, however exons 16 and 21, at least in our cohort of patients, appeared to be more mutation rich.

NF1 is a multisystem disorder associated to several clinical complications, including skeletal abnormalities. The spine deformities were manifested in the 75% of our case series and scoliosis was the most common musculoskeletal manifestation (50/85 patients), in line with past research [107,108]. In the 33% of patients, the skeletal phenotype was worsened by the concomitant expression of both spine and bone deformities. In our cohort, patients carrying deletion of the entire *NF1* gene showed spine abnormalities, and the most common mutations associated to this clinical trait resulted in truncated neurofibromin. NF_40 patient, showing kyphoscoliosis, carried the same pathogenic mutation c.1466A>G, p.Tyr489Cys previously identified in four NF1 patients with scoliosis [42]. The 21% of NF1 patients manifested spinal neurofibromas that were strongly associated with pain and scoliosis. Learning disabilities, including visual perception, language, executive functions, attention, and motor skills, represent one of the most common and challenging complications of NF1 [109]. Previous research reported the association of facial dysmorphism and learning deficits with *NF1* microdeletions [27].

Interestingly, our results demonstrated a higher frequency of cognitive disorder, including delayed acquisition of psychomotor skills and residual multi-domain cognitive impairment, in patients carrying *NF1* frameshift mutations (OR 6.2, CIs 1.626 to 23.64, *p* < 0.01). To support these preliminary findings further investigations are needed. The involvement of the cardiovascular system in NF1 is strongly supported by several studies [110,111,112]. Cardiovascular disorders were detected in 28% of our cohort. Two NF1 patients, NF_10 affected by hypertension while NF_11 showed mitral and aortic insufficiency, carried the nonsense mutation, c.3826C>T, p.Arg1276* that was previously described to be correlated with a high prevalence of cardiovascular abnormalities [34].

### NF1 and Cancer Implication

Individuals affected by NF1 present a higher risk to develop specific malignancies compared to the general population [113]. A wide range of benign and malignant tumors in both central and peripheral nervous systems, as well as other organ malignancies, has been described in association to NF1 [113].

In this study, nervous system malignancies and organ tumors affected respectively 21% and 22% of our cohort. PNFs represent an uncommon variant (5–15%) of NF1 [114]. According to scientific literature, in our series PNFs represented a rare clinical condition (13%; 11/85 patients). MPNSTs are the most common malignant tumors associated with NF1 condition, usually arising from PNFs, and represent the major cause of poor prognosis [11,115]. In our study, we described three NF1 patients, belonging to G3 clinical group, affected by MPNST. NF_18 presented lumbar spinal MPNST associated to c.3326T>G, p.Leu1109* mutation, located in the tubulin-binding domain (TBD) of NF1 protein. Recently, this mutation has been described in one NF1 patient showing mild phenotype, without MPNST [36]. NF_52 carried the novel c.4309G>T, p.(Glu1437*) mutation in the GRD and manifested MPNST of the left thigh, in combination to GIST. NF_127 was affected by a rare condition of intracranial MPNST linked to c.2540T>G, p.Leu847Arg missense mutation in the cysteine and serine-rich domain (CSRD). It was interesting that this mutation was described in nine NF1 patients with a very high number of neurofibromas, including two individuals with metastasized MPNSTs [30]. Optic glioma (OPG) and astrocytoma were the most NF1-associated nervous system malignancies. Recent evidence has suggested that the specific genotype may be the main determinant of the development of OPG, with the risk being higher in patients harboring *NF1* mutations in the 5’ tertile (exon 1–21) and in the CSRD domain (residues 543–909), whereas mutations in the HEAT-like repeat regions (HLR, 1825–2428) were negatively associated with OPG [116,117,118]. We described ten NF1 patients showing OPG, two of which presented OPG and astrocytoma. In agreement to recent evidence, four patients (NF_26, 60, 76, 115) carried mutations in the 5’ tertile and in CSRD domain. Nevertheless, further investigation of larger case series of patients will be needed to confirm our preliminary evidence. Other malignancies detected in our case series include GIST, breast and prostate cancers, seminoma, lymphoma and malignant pheochromocytoma. In our cohort three different nonsense mutations c.3826C>T, p.Arg1276* in NF_19; c.4923G>A, p.Trp1641* in NF_73; c.7089dup, p.Asn2364* in NF_98 were associated to malignant pheochromocytoma.

It is intriguing that the p.Arg1276* mutation was associated to cancer development (our study and ref. [77]) and cardiovascular abnormalities (our study in NF_10, 11, and ref. [34]); possibly a different genetic background (e.g., variants in NF1-related genes) or non-genetic factors (e.g., environment, exposure to chemicals and hazardous substances, etc.) may explain the diverse clinical manifestations.

The NF_92 patient, carrying the c.4278G>C, p.Gln1426His missense mutation, showed GIST, lymphoma associated to malar facial PNF as well as spine deformities and aortic and mitral valves insufficiency. This mutation was previously reported associated to NF1 with pulmonary stenosis [78] and in two NF1 patients with mild phenotype [36]. Moreover, in our study five different *NF1* mutations were shared in unrelated patients, showing different clinical manifestations.

Furthermore, findings of segregation analysis in the 20 familial cases of our case series confirmed the variability in clinical expression of NF1 within relatives carrying the same *NF1* mutations. A typical case is the NF_29 patient, that, although sharing the same missense mutation c.1595T>G, p.Leu532Arg with the sibling NF_5, had a more severe phenotype worsened by epilepsy syndrome and seminoma (NF_5, G2; NF_29, G5).

On the other hand, the influence of variants in other genes on NF1 phenotype cannot be excluded. Several studies showed that different mutations of *NF1* and related genetic modifiers might contribute together to clinical features in NF1, including tumor development, making the scenario more complex [119]. It will be very important to perform this analysis in our cohort of patients in the future.

Finally, a limitation of the NF1 cohort study we present may be related to the small size of each patient group (G1-G5). However, given the considerable clinical heterogeneity of NF1 patients, the inclusion of phenotypically homogeneous NF1 individuals in each group allowed a more appropriate genotype–phenotype analysis and more reliable conclusions. Undoubtedly, the observation of a larger cohort of NF1 patients will allow further considerations of clinical-genetic utility.

## 5. Conclusions

The cohort of NF1 patients presented here provides the opportunity to expand the spectrum of *NF1* mutations and confirms that NF1 is a highly phenotypically heterogeneous disease with a wide range of effects on the CNS and consequent impairment of cognitive and educational functions. Indeed, some NF1 patients present with an uncomplicated picture, which is the most common case, but others may develop serious manifestations, such as tumors of different organs and apparatuses and of varying malignancy, which may complicate the clinical picture.

Studies looking at specific cognitive domains associated with NF1 have come to very different conclusions. In our cohort, we found a higher prevalence of learning disabilities in patients with *NF1* frameshift mutations. Findings from several studies involving very young NF1 children confirm that developmental delays and subsequent academic difficulties and learning disabilities represent a major psychosocial burden during the life of NF1 patients. In addition, our study particularly highlights that cognitive aspects, such as the delay in the acquisition of psychomotor skills, unlike other symptoms, can be a warning sign, allowing for earlier diagnosis and a more effective therapeutic and rehabilitation approach to reduce residual cognitive impairment in multiple domains and the psychosocial impact of the disease. Hence, the perception of disease severity correlates with medical, behavioral, and cognitive severity scores, and the spectre of death from cancer or other complications has profound emotional, psychological, and social effects on all NF1 patients and their families. Therefore, for adequate genotype–phenotype analysis in NF1, better clinical management and appropriate genetic counseling we advocate a longitudinal, patient-centered model of care, with age-dependent monitoring of clinical manifestations aimed at early diagnosis and symptomatic treatment of emerging complications.

Overall, our study aims to contribute to a better definition of the genotype–phenotype correlation and may improve the management and genetic counseling of NF1 patients.

## Figures and Tables

**Figure 1 genes-13-01130-f001:**
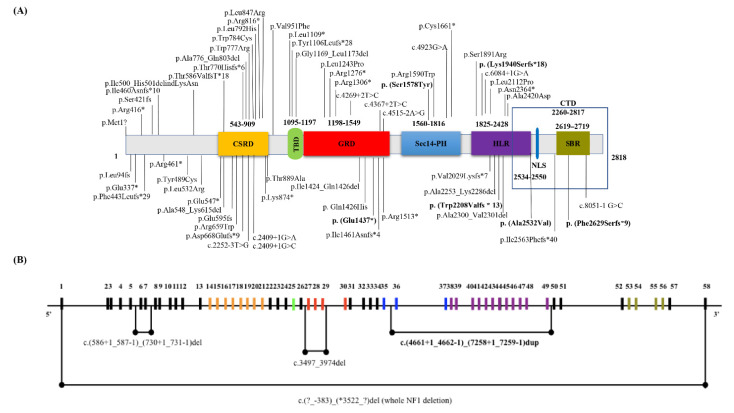
(**A**) **Distribution of the identified mutations in the *NF1* gene.** Details of the 66 *NF1* genetic mutations identified in our NF1 Italian cohort. The position of genetic variations detected in the *NF1* from each NF1 patient is shown and their distribution in the NF1 domains is reported. Vertical lines show variant position. *NF1* novel variants are shown in bold. Neurofibromin domains: CSRD, cysteine/serine-rich domain (543–909 residues); GRD, GAP related domain (1198–1549 residues); TBD, tubulin-binding domain (1095–1197 residues); HLR, HEAT-like repeat regions (1825–2428 residues); NLS, nuclear localization signal (2534–2550 residues); Sec14-PH, Sec14-homologous domain (1560–1816 residues) and Pleckstrin Homology domain (1716–1816 residues); CTD, C-terminal domain (2260–2817 residues); SBR, Syndecan-Binding Region (2619–2719 residues). (**B**) **Map of the NF1 region indicating the large duplications and deletions sequences identified in our NF1 cohort.** The black horizontal line represents the entire *NF1* gene, while the vertical lines indicate the *NF1* exons. Different colors were used for *NF1* exons according to the relative functional domains as shown in (**A**).

**Figure 2 genes-13-01130-f002:**
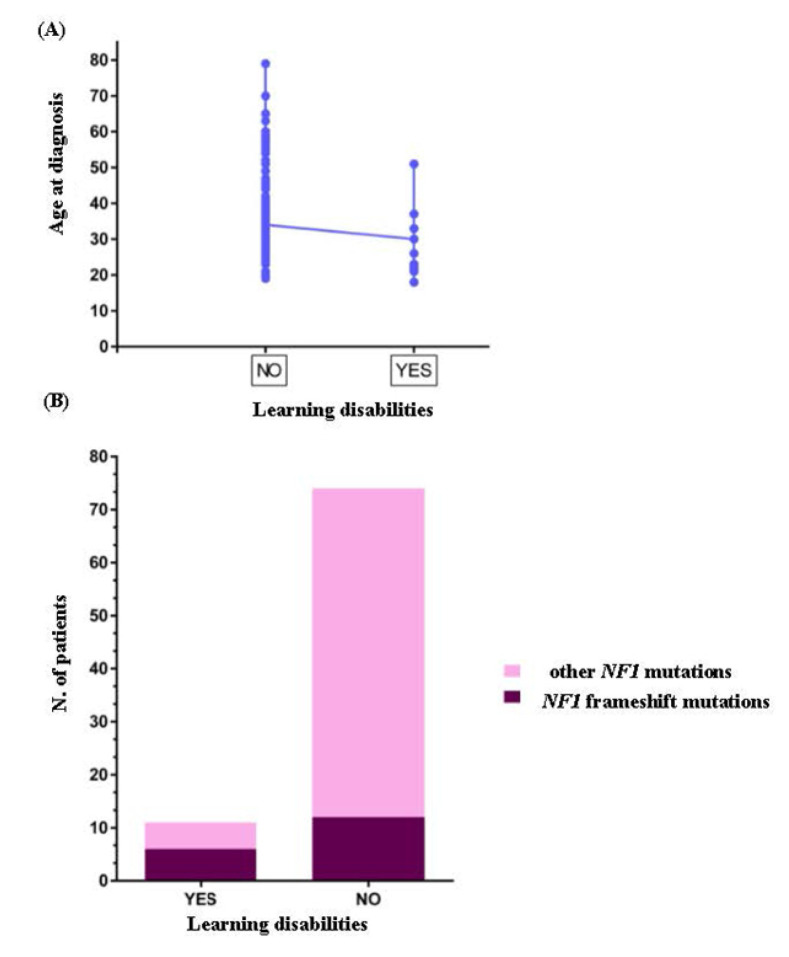
(**A**) Correlation analysis in the NF1 patients. Graph showing an inverse correlation between the presence of learning disabilities and age at diagnosis. (**B**) Odds ratio analysis. Odds ratio with 95% CI revealed a higher probability to have learning disabilities in NF1 patients carrying frameshift mutation in the *NF1* gene.

**Table 1 genes-13-01130-t001:** Molecular characteristics of the NF1 cohort, according to clinical groups (G1–G5).

ID	Clinical Group	Nucleotide Change	Amino Acid Change	Exon/ Intron	Mutation Type	CS	Ref.	FH	ClinVar/ HGMD/ LOVD	PD
**NF_2**	G1	c.3G>A	p.?	1	Start lost	P	[39,40]	Pat	LOVD: NF1_001130	-
**NF_54**	G1	c.1639G>T	p.Glu547*	14	Nonsense	P	[39]	-	LOVD: NF1_001185	CSRD
**NF_68**	G1	c. 6792C>A	p.Ala2253_Lys2286del	46	Splicing affected	P	[41,42,43]	-	LOVD: NF1_000816	HLR
**NF_83**	G1	c.5673T>G	p.Ser1891Arg	39	Missense	-	-	-	LOVD: NF1_002964	HLR
**NF_89**	G1	c.(4661+1_4662-1)_(7258+1_7259-1)dup	-	-	Duplication	-	-	-	NR	Sec14-PH /HLR
**NF_1**	G2	c.(586+1_587-1)_(730+1_731-1)del	-	6,7	Deletion	P	[44,45]	Unk	-	-
**NF_3**	G2	c.3G>A	p.?	1	Start lost	P	[39,40]	Unk	LOVD: NF1_001130	-
**NF_4**	G2	c.3G>A	p.?	1	Start lost	P	[39,40]	Pat	LOVD: NF1_001130	-
**NF_5**	G2	c.1595T>G	p.Leu532Arg	14	Missense	P	[36,40]	Pat	LOVD: NF1_002498	-
**NF_8**	G2	c.7884_7885del	p.(Phe2629Serfs*9)	-	Frameshift	-	-	-	NR	SBR
**NF_10**	G2	c.3826C>T	p.Arg1276*	28	Nonsense	P	[34,46]	Mat	ClinVar: variation ID 237556	GRD
**NF_11**	G2	c.3826C>T	p.Arg1276*	28	Nonsense	P	[34,46]	Unk	ClinVar: variation ID 237556	GRD
**NF_13**	G2	c.6892_6897del	p.Ala2300_Val2301del	-	Deletion	-	[36]	-	-	HLR
**NF_20**	G2	c.1783_1784del	p.Glu595fs	16	Frameshift	P	[39]	-	LOVD: NF1_001194	CSRD
**NF_21**	G2	c.(?_-383)_(*3522_?)del (whole *NF1* deletion)	-	-	Deletion	P	[27,47,48,49,50,51]	Mat	-	-
**NF_25**	G2	c.3496+1G>A	p.Tyr1106Leufs*28	26	Frameshift	-	[36]	Mat	HGMD: CS072245	TBD
**NF_28**	G2	c.1A>G	p.?	1	Missense	P	[52,53,54]	Pat	LOVD: NF1_000140	-
**NF_30**	G2	c.3728T>C	p.Leu1243Pro	28	Missense	LP	[36]	Pat	LOVD: NF1_001544	GRD
**NF_38**	G2	c.4768C>T	p.Arg1590Trp	36	Missense	VOUS	[36,55,56,57,58,59]	-	HGMD: CM971051	Sec14-PH
**NF_40**	G2	c.1466A>G	p.Tyr489Cys	13	Splicing affected	P	[41,54,60,61,62,63]	Pat	LOVD: NF1_000063	-
**NF_43**	G2	c.2352G>C	p.Trp784Cys	20	Missense	P	[36,64,65]	Pat	LOVD: NF1_001853	CSRD
**NF_45**	G2	c.2307dup	p.Thr770Hisfs*6	19	Frameshift	-	[36]	Pat	-	CSRD
**NF_46**	G2	c.2307dup	p.Thr770Hisfs*6	19	Frameshift	-	[36]	Pat	-	CSRD
**NF_47**	G2	c.2307dup	p.Thr770Hisfs*6	19	Frameshift	-	[36]	Pat	-	CSRD
**NF_48**	G2	c.1A>G	p.?	1	Missense	P	[52,53,54]	-	LOVD: NF1_000140	-
**NF_50**	G2	c.3497_3974del	-	-	Deletion	-	[36,54,66,67]	-	-	GRD
**NF_51**	G2	c.4381dup	p.Ile1461Asnfs*4	34	Frameshift	P	[68]	Pat	LOVD: NF1_001553	GRD
**NF_53**	G2	c.1378dup	p.Ile460Asnfs*10	12	Frameshift	-	[36]	-	-	-
**NF_56**	G2	c.2409+1G> Ac.2375T>A	p.? p.Leu792His	20i	Splicing affected Missense	P LP	[62,69] -	-	LOVD: NF1_000203 ClinVar: variation ID 665425	- CSRD
**NF_57**	G2	c.1381C>T	p.Arg461*	12	Nonsense	P	[42,53,61,70,71,72]	-	LOVD: NF1_000056	-
**NF_59**	G2	c.4270-2A>G	p.Ile1424_Gln1426del	32i	Splicing affected	P	[73,74]	Mat	LOVD: NF1_000479	GRD
**NF_61**	G2	4367+2T>C	p.?	-	Splicing affected	LP	-	Pat	ClinVar: variation ID 527560	-
**NF_62**	G2	c.1260+1G>A	p.Ser421fs	11i	Splicing affected	P	[75,76]	Pat	LOVD: NF1_000036	-
**NF_63**	G2	c.4733C>A	p.(Ser1578Tyr)	-	Missense	-	-	Unk	NR	Ses14-PH
**NF_66**	G2	c.2665 A>G	p.Thr889Ala	-	Missense	VOUS	-	-	ClinVar: variation ID 527580	CSRD
**NF_71**	G2	c.2409+1G>C	p.?	20i	Splicing affected	P	[69,77]	Pat	LOVD: NF1_000204	-
**NF_77**	G2	c.2326G>A	p.Ala776_Gln803del	-	Splicing affected	-	[36]	-	-	CSRD
**NF_80**	G2	c.4278G>C	p. Gln1426His	33	Missense	P	[36,78]	Pat	ClinVar: variation ID 233115	GRD
**NF_86**	G2	c.4923G>A	p.Trp1641*	-	Nonsense	P	[52]	Pat	LOVD: NF1_001303	Sec14-PH
**NF_90**	G2	c.(4661+1_4662-1)_(7258+1_7259-1)dup	-	-	Duplication	-	-	Mat	NR	Sec14-PH /HLR
**NF_94**	G2	c.2252-3T>G	P.?	-	Splicing affected	P	[79]	-	ClinVar: variation ID 374022	-
**NF_96**	G2	c.(?_-383)_(*3522_?)del (whole *NF1* deletion)	-	-	-	P	[27,47,48,49,50,51]	-	-	-
**NF_100**	G2	c.4537C>T	p.Arg1513*	35	Nonsense	P	[42,63,80,81,82,83,84]	-	LOVD: NF1_000521	GRD
**NF_102**	G2	c.1381C>T	p.Arg461*	12	Nonsense	P	[41,42,53,61,70,71,72]	Pat	LOVD: NF1_000056	-
**NF_103**	G2	c.2251 G>C	p.Asp668Glufs*9	-	Frameshift	-	[36]	+	-	CSRD
**NF_18**	G3	c.3326T>G	p.Leu1109*	26	Nonsense	-	[36]	-	-	TBD
**NF_52**	G3	c.4309G>T	p.(Glu1437*)	-	Nonsense	-	-	-	NR	GRD
**NF_127**	G3	c.2540T>G	p.Leu847Arg	21	Missense	P	[24,30,38,53,61,65,70,85,86,87,88]	-	ClinVar: variation ID 573019	CSRD
**NF_6**	G4	c.479+5G>A	p.Leu94fs	4i	Frameshift	P	[55,89]	Mat	ClinVar: variation ID 237521	-
**NF_17**	G4	c.6364+4A>G	p.Val2029Lysfs*7	41i	Splicing affected	-	[36]	-	HGMD: CS941517	HLR
**NF_26**	G4	c.1246C>T	p.Arg416*	11	Nonsense	P	[70,90]	-	LOVD: NF1_000034	-
**NF_35**	G4	c.4269+2T>C	p.?	32i	Splicing affected	P	[36]	Pat	-	-
**NF_36**	G4	c.6335T>C	p.Leu2112Pro	42	Missense	P	[36]	-	LOVD: NF1_000756	HLR
**NF_37**	G4	c.(?_-383)_(*3522_?)del (whole *NF1* deletion)	-	-	Deletion	P	[27,47,48,49,50,51]	-	-	-
**NF_41**	G4	c.1499_1501delinsAAA	p.Ile500_His501delinsLysAsn	13	INDEL	-	[36]	-	-	-
**NF_60**	G4	c.1756_1759del	p.Thr586Valfs*18	16	Frameshift	P	[42,62,91,92,93,94,95,96]	Pat	LOVD: NF1_000113	CSRD
**NF_65**	G4	c.(?_-383)_(*3522_?)del (whole *NF1* deletion)	-	-	Deletion	P	[27,47,48,49,50,51]	Mat	-	-
**NF_67**	G4	c.6084+1G>A	p.?	-	Splicing affected	P	[38,39,43,70,71,97]	Pat	ClinVar: variation ID 404489	-
**NF_70**	G4	c.2409+1G>C	p.?	20i	Splicing affected	P	[69,77]	Pat	LOVD: NF1_000204	-
**NF_72**	G4	c.7884_7885del	p.(Phe2629Serfs*9)	-	Frameshift	-	-	Pat	NR	SBR
**NF_76**	G4	c.1845+1_1845+5del	p.Ala548_Lys615del	16	Splicing affected	P	[68,70]	-	LOVD: NF1_001511	CSRD
**NF_81**	G4	c.5819del	p.(Lys1940Serfs*18)	-	Frameshift	-	-	Unk	NR	HLR
**NF_82**	G4	c.8051-1 G>C	p.?	-	Splicing affected	-	[36]	-	-	-
**NF_85**	G4	c.6621dup	p.(Trp2208Valfs*13)	-	Frameshift	-	-	-	NR	HLR
**NF_99**	G4	c.2851G>T	p.Val951Phe	22	Missense	LP	[68]	Mat	LOVD: NF1_001526	-
**NF_115**	G4	c.2446C>T	p.Arg816*	21	Nonsense	P	[56,70,98]	-	LOVD: NF1_000214	CSRD
**NF_15**	G5	c.2619dup	p.Lys874*	21	Nonsense	P	[41,60]	-	ClinVar: variation ID 404563	CSRD
**NF_19**	G5	c.3826C>T	p.Arg1276*	28	Nonsense	P	[34,46]	Mat	LOVD: NF1_000403	GRD
**NF_22**	G5	c.4982_4983del	p.Cys1661*	37	Nonsense	P	-	-	LOVD: NF1_000602	Sec14-PH
**NF_23**	G5	c.3496+1G>A	p.Tyr1106Leufs*28	26	Frameshift	-	[36]	Unk	HGMD: CS072245	TBD
**NF_29**	G5	c.1595T>G	p.Leu532Arg	14	Missense	P	[36,40]	Pat	LOVD: NF1_002498	-
**NF_31**	G5	c.7686delG	p.Ile2563Phefs*40	53	Frameshift	P	[36,99]	Pat	LOVD: NF1_002529	CTD
**NF_32**	G5	c.7686delG	p.Ile2563Phefs*40	53	Frameshift	P	[36,99]	Pat	LOVD: NF1_002529	CTD
**NF_39**	G5	c.1009G>T	p.Glu337*	-	Nonsense	-	-	Pat	ClinVar: 439994	-
**NF_42**	G5	c.3502-3519del	p.Gly1169-Leu1173del	-	-	-	[36]	Mat	-	TBD
**NF_55**	G5	c.4537C>T	p.Arg1513*	35	Nonsense	P	[41,42,63,80,81,82,83,84]	-	LOVD: NF1_000521	GRD
**NF_73**	G5	c.4923G>A	p.Trp1641*	37	Nonsense	P	[52]	Unk	LOVD: NF1_001303	Sec14-PH
**NF_78**	G5	c.2329T>C	p.Trp777Arg	20	Missense	P	[1,39,100]	Mat	LOVD: NF1:000186	CSRD
**NF_79**	G5	c.3916C>T c.1975C>T	p.Arg1306* p.Arg659Trp	29 17	Nonsense Missense	P VOUS	[41,54,67,70,91,101,102][98]	-	LOVD: NF1_000416 LOVD: NF1_002592	GRD CSRD
**NF_87**	G5	c.7259C>A	p.Ala2420Asp	50	Missense	-	-	-	LOVD: NF1_000867	HLR
**NF_92**	G5	c.4278G>C	p.Gln1426His	33	Missense	P	[36,78]	Pat	ClinVar: Variation ID 233115	GRD
**NF_97**	G5	c.4515-2A>G	p.?	34i	Splicing affected	P	-	Mat	LOVD: NF1_000518	-
**NF_98**	G5	c.7089dup	p.Asn2364*	48	Nonsense	-	-	-	LOVD: NF1_001359	HLR
**NF_101**	G5	c.1329delT	p.Phe443Leufs*29	-	Frameshift	-	[36,103,104]	-	-	-
**NF_105**	G5	c.7532C>T	p.(Ala2532Val)	-	Missense	-	-	Mat	NR	CTD

## Data Availability

Due to the sensitive nature of the data, information created during and/or analysed during the current study is available from corresponding authors on reasonable request.

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
