# Peer review of "Genotype-Phenotype Correlations in Neurofibromatosis Type 1: Identification of Novel and Recurrent NF1 Gene Variants and Correlations with Neurocognitive Phenotype"

_genes, 2022, doi:10.3390/genes13071130_

Round 1
Reviewer 1 Report
The authors conducted an interesting study describing neurofibromatosis type 1 mutational spectrum using a robust methodology and genotype-phenotype correlations. The description of novel variants and an attempt to establish genotype-phenotype correlations in Neurofibromatosis type 1 is relevant and very important to understand disease variability and offer better genetic counseling to the patients. I have minor comments to the authors:
- I think that the introduction section is long and could be more focused on NF1 genotype-phenotype correlations.
- The authors stated that “The classification was conducted in accordance with the guidelines from the American College of Medical Genetics and Genomics”. However, this guideline does not have a category named “unknown significance”. Additionally, please provide reference and version of the American College of Medical Genetics and Genomics guidelines used.
- Please provide more details about statistical analysis – patients with the same type of mutations were grouped in each of the clinical groups (G1, G2..)?
- In the results section, please provide the average depth, depth of the identified variants and minimum and maximum depth obtained in NGS in each sample.
- The authors could briefly discuss the influence of variants in other genes in NF1 phenotype, which has been investigated in scientific studies.
- The authors could discuss the limitations of their study, including the small number of patients in each clinical group.
Reviewer 2 Report
Please see the attached review.

Round 2
Reviewer 2 Report
In my opinion the Authors significantly improved the manuscript. However, there are still some issues that need to be addressed.
Please see the attached review.
